# HIV-1-Infected Human Macrophages, by Secreting RANK-L, Contribute to Enhanced Osteoclast Recruitment

**DOI:** 10.3390/ijms21093154

**Published:** 2020-04-30

**Authors:** Rémi Mascarau, Florent Bertrand, Arnaud Labrousse, Isabelle Gennero, Renaud Poincloux, Isabelle Maridonneau-Parini, Brigitte Raynaud-Messina, Christel Vérollet

**Affiliations:** 1Institut de Pharmacologie et Biologie Structurale, Université de Toulouse, CNRS UMR 5089, Université Toulouse III Paul Sabatier, CEDEX 04, 31077 Toulouse, France; remi.mascarau@ipbs.fr (R.M.); bertrandflorent.bf@gmail.com (F.B.); arnaud.labrousse@ipbs.fr (A.L.); poincloux@ipbs.fr (R.P.); maridono@ipbs.fr (I.M.-P.); 2Centre de Physiopathologie de Toulouse-Purpan, INSERM-CNRS UMR 1043, Université Toulouse III Paul Sabatier, 31024 Toulouse, France; gennero.i@chu-toulouse.fr; 3Institut Fédératif de Biologie, Centre Hospitalier Universitaire Toulouse, 31059 Toulouse, France; 4International Associated Laboratory (LIA) CNRS “IM-TB/HIV” (1167), 31077 Toulouse, France; 5International Associated Laboratory (LIA) CNRS “IM-TB/HIV” (1167), Buenos Aires C1425AUM, Argentina

**Keywords:** HIV-1, macrophages, osteoclasts, bone defects, RANK-L, cell migration

## Abstract

HIV-1 infection is frequently associated with low bone density, which can progress to osteoporosis leading to a high risk of fractures. Only a few mechanisms have been proposed to explain the enhanced osteolysis in the context of HIV-1 infection. As macrophages are involved in bone homeostasis and are critical host cells for HIV-1, we asked whether HIV-1-infected macrophages could participate in bone degradation. Upon infection, human macrophages acquired some osteoclast features: they became multinucleated, upregulated the osteoclast markers RhoE and β3 integrin, and organized their podosomes as ring superstructures resembling osteoclast sealing zones. However, HIV-1-infected macrophages were not fully differentiated in osteoclasts as they did not upregulate NFATc-1 transcription factor and were unable to degrade bone. Investigating whether infected macrophages participate indirectly to virus-induced osteolysis, we showed that they produce RANK-L, the key osteoclastogenic cytokine. RANK-L secreted by HIV-1-infected macrophages was not sufficient to stimulate multinucleation, but promoted the protease-dependent migration of osteoclast precursors. In conclusion, we propose that, by stimulating RANK-L secretion, HIV-1-infected macrophages contribute to create a microenvironment that favors the recruitment of osteoclasts, participating in bone disorders observed in HIV-1 infected patients.

## 1. Introduction

Low bone density is frequent in Human Immunodeficiency Virus type 1 (HIV-1) infected patients and can progress to osteoporosis and high risk of fractures [1]. While lifespan of patients has significantly increased with antiretroviral therapy, long-term complications such as bone defects appeared. Multiple factors such as antiretroviral therapy and patient life style and health (e.g., body mass index, age or muscle wasting), contribute to bone loss in infected patients [2]. Moreover, bone deficits in non-treated patients attest for a role of the virus by-itself [3,4,5,6].

The skeleton is a dynamic organ undergoing continual remodeling owing to the actions of bone-resorbing osteoclasts (OC), bone-forming osteoblasts and osteocytes. The balance between osteoblasts and OC can be disturbed leading to bone defects. Some in vitro studies reported effects of viral proteins (e.g., gp120, Tat and Nef) on bone cells including osteoblasts [7,8,9]. However, in HIV-1^+^ patients, bone loss is associated with an increase in blood biomarkers of bone resorption but no or slight modification of markers of bone formation, suggesting a major contribution of OC [6]. While osteoblasts originate from cells of a mesenchymal origin, OC mainly develop by fusion of monocytic precursors derived from hematopoietic stem cells [10,11]. OC are multinucleated giant cells that differentiate from fusion of myeloid precursors under the control of Macrophage Colony-Stimulating Factor (M-CSF) and Receptor Activator of Nuclear Factor-κB Ligand (RANK-L) [10,11,12]. The degree of OC differentiation depends mainly on the activity of RANK-L that is moderated by its physiological decoy receptor osteoprotegerin (OPG) [13]. OC differentiation mainly occurs through activation of the transcription factor Nuclear Factor of Activated T cells cytoplasmic 1 (NFATc1) [14]. In addition, proinflammatory cytokines such as Tumor Necrosis Factor (TNF)-α, InterLeukin (IL)-6, IL-1β, and IL-17 favor osteoclastogenesis, i.e. recruitment and/or differentiation of OC precursors, and thus provide a supportive environment for osteoclastogenesis [15]. Terminally differentiated OC express high levels of the αvβ3 integrin adhesion receptor, Tartrate Resistant Acidic Phosphatase (TRAP) and resorption-related enzymes, including Cathepsin K (Ctsk), and the C1 subunit of the V-type proton ATPase (ATP6v1c1). OC attachment to bone is mediated by an OC-specific structure called sealing zone. It is composed of a dense array of inter-connected F-actin structures, called podosomes that form a circular superstructure. Podosomes are present mainly in myeloid cells, including macrophages (MF), dendritic cells, and OC [16]. The sealing zone anchors OC to the bone surface and creates a confined resorption environment, where protons and osteolytic enzymes are secreted [12,14]. The formation of this structure is strongly regulated, involving the Src tyrosine kinase [17] and the small GTPase RhoE [18].

Today, only a few mechanisms have been proposed to explain the increase in osteolytic activity associated with HIV-1 infection comprising increased production of proinflammatory cytokines [19], disruption of the immune system [6,20,21,22], and activation of the bone resorption activity of infected OC [23,24,25]. Through its action on T and B cells, HIV-1 infection leads to an increase in the RANK-L/OPG ratio that stimulates OC differentiation [6,21,22]. In addition, we have shown that HIV-1 infects OC precursors, enhancing their migration to bones and differentiation, but also mature OC, inducing modifications in the structure and function of the sealing zone. These changes markedly correlate with an enhanced OC adhesion and bone degradation capacity. This exacerbated osteolytic activity of infected OC is dependent on Src activation by the viral protein Nef [25]. 

Along with CD4 T lymphocytes, MF are critical host cells for HIV-1. Recent data highlight the capacity of MF to sustain active viral replication in vivo, their resistance to the viral cytopathic effects and their distribution in most tissues of the organism. MF can serve both as a vehicle for viral transmission, dissemination, and cellular reservoir [26,27,28,29,30,31]. Moreover, MF plays an important role in bone homeostasis and repair, involving a collaboration between infiltrating monocyte-derived MF and resident MF (also called osteal MF or osteomacs) [32]. Bone MF differs from OC in their specific expression of Siglec1 (CD169) [33]. They are in close contact with osteoblasts, thus participating in the regulation of bone mineralization. They play a critical role in immune responses to pathogens and to biomaterials, and also contribute to the induction, progression, and resolution of fracture repair [32,34,35].

Thus, in order to further understand the mechanisms involved in the bone defects induced by HIV-1 infection, we challenge a novel hypothesis that proposes a role for HIV-1-infected MF in virus-induced osteolysis. We examined whether HIV-1 infection would induce a reprogramming of MF toward an OC signature. Several arguments support this proposition: upon inflammatory conditions, myeloid cells such as dendritic cells can transdifferentiate into functional OC [36,37,38] and in vitro, HIV-1-infected MF acquire some OC characteristics (multinucleation, enhanced capacity to degrade organic matrices, and when seeded on glass, organization of their podosomes into circular structures) [30,39]. Another possibility would be an indirect effect of MF infection *via* a modification in secreted molecules, which may exert bystander effects on the recruitment or the differentiation of surrounding OC precursors. Here, we obtained evidence on the critical role of HIV-infected MF on the secretion of RANK-L, which supports osteoclastogenesis.

## 2. Results

### 2.1. HIV-1 Infection of Macrophages Induces the Expression of a Subset of Osteoclast Markers

To determine whether HIV-1-infected MF acquired OC characteristics, human MF derived from primary monocytes were infected with two R5 MF-tropic strains, ADA and NLAD8. Ten days post-infection, both strains induced an efficient and productive infection as judged by the percentage of infected cells determined by immunofluorescence (IF) analysis with an antibody against the viral protein gag (Figure 1A), by quantification of the intracellular mRNA level and the concentration of gag in cell supernatants (Appendix A). We quantified the fusion index, which corresponds to the percentage of nuclei within multinucleated giant cells (MGC), from large IF images (Figure 1B). As expected from previous data [39,40], HIV-1 infection of MF with either ADA or NLAD8 strains triggered efficient MF fusion into MGC compared to non-infected cells. Interestingly, in our experimental conditions, the number of nuclei per MGC was higher in HIV-1-infected MF (Nb of nuclei/cell; 2–4 nuclei: 65 ± 2%, 5–9 nuclei: 26 ± 1%, >10 nuclei: 9 ± 1%, *n* = 5 donors) compared to OC (Nb of nuclei/cell; 2–4 nuclei: 86 ± 5%, 5–9 nuclei: 12 ± 3%, >10 nuclei: 2 ± 1%, *n* = 5 donors). However, the number of MGC was higher in OC compared to HIV-infected MF, resulting in a similar fusion index for the two populations (Figure 1A,B). MGC formation was also observed after infection of MF with two clinical viral strains [41] (Appendix A), even if the infectivity rate was lower. Then, we analyzed the expression of genes, which are specifically overexpressed during OC differentiation (mRNA level quantification normalized to non-infected MF differentiated from the same donor). We first verified that the genes encoding for the transcription factor NFATc1, the osteolytic enzymes TRAP, Ctsk, and ATP6v1c1 as well as the β3 integrin subunit and RhoE were induced in OC compared to non-infected MF. In HIV-1-infected MF (ADA strain) compared to uninfected MF, the level of integrin β3 and RhoE mRNA was increased by two-fold, whereas the one of Nfatc1, TRAP, Ctsk, and ATP6v1c1 was not increased (Figure 1C). Although not significant, similar results were obtained with MF infected with NLAD8 strain. Western blot analyses showed that the variations in mRNA expression level translated into increased protein expression of β3 integrin and RhoE in MF infected with both viral strains compared to non-infected MF (Figure 1D). Consistently with RT-qPCR analysis, no change was observed in the expression level of Ctsk protein (Appendix A). 

Together, these results indicate that HIV-1 infection of MF induces multinucleation and the expression of OC cytoskeletal markers, i.e., RhoE and β3 integrin, but is not sufficient to trigger the expression of all characteristic genes/proteins of OC.

### 2.2. HIV-1 Infection of Macrophages Induces the Formation of Sealing Zone-Like Structures 

As HIV-1-infected MF showed a partial differentiation toward OC, we next characterized the consequences of infection on the organization of podosomes, the main F-actin structure in MF and OC [16]. When cells were seeded on glass, most uninfected MF exhibited scattered podosomes while the majority of HIV-1(NLAD8)-infected MF reorganized their podosomes into clusters, rings, or belts, as described for OC at different stages of differentiation [42] (Figure 2A). Similar results were obtained upon infection of MF with the ADA strain [39]. The capacity of HIV-1-infected MF to form actin circular structures on glass was, at least, equivalent to the one of OC from the same donor. When cells were seeded on bone slices, we observed that about 15% of HIV-1 infected MGC formed podosome super-structures whose size was systematically smaller than the one of OC sealing zone (Figure 2B,C). Non-infected MF did not present any specific actin organization while 70% of OC formed a classical sealing zone (Figure 2B,C and Appendix A). When looking at vinculin, a protein accumulated in podosomes, we showed that this protein colocalized to the actin ring structure assembled in HIV-1-infected MF, as observed in OC sealing zone (Figure 2D). 

Thus, HIV-1-infected MF forms F-actin structures sharing common characteristics with the OC sealing zone, even if their number and size are lower.

### 2.3. HIV-1 Infection of Macrophages Does not Enhance Their Bone Degradation Activity 

As the formation of sealing zone is instrumental for osteolytic activity, we explored the capacity of HIV-1-infected MF to degrade the bone matrix. The bone resorption area observed for MF (close to 1% of the total bone surface) did not significantly increase upon HIV-infection (Appendix A), while it reached 15% for OC differentiated from the same donors. This parameter correlated with a low concentration of the C-terminal telopeptide of type 1 collagen (CTX) released in the supernatants of uninfected and infected MF, compared to the one obtained in OC supernatants which was around 50-fold higher (Appendix A). The morphology of resorption lacunae was examined by scanning electron microscopy (SEM). As expected, OC formed long and large bone degradation trails [14,25,43] (Figure 3, upper panels). In the case of infected MF, we observed in some experiments (3 donors out of 5) the rare appearance of individual round resorption pits referred to short trails (Figure 3, lower panels); this degradation profile was never observed for uninfected MF. High magnification of the degraded pits showed superficial abrasion of bone microstructure in the case of infected MF compared to deep excavations with high porosity in the case of OC. 

All these data show that while HIV-1-infected MF acquire some OC characteristics such as multinuclearity and organized podosome rings on bones, they are not fully competent for bone resorption.

### 2.4. RANK-L Is Secreted by Infected Macrophages and Promotes 3D Migration of Osteoclast Precursors 

We then investigated whether HIV-infected MF, through a modification in secreted molecules, may exert bystander effects on the recruitment or differentiation of surrounding OC precursors. Indeed, an abundant literature reports the importance of cytokines in osteoclastogenesis, as they control both chemotaxis of OC precursors and OC differentiation *per se* [15]. First, the secretion of the key osteoclastogenic cytokine RANK-L was examined. Interestingly, in conditioned medium of HIV-1-infected MF (CmHIV), RANK-L concentration was increased in comparison with conditioned medium of uninfected MF (CmNI) (300 pg/mL of RANK-L after infection *versus* 70 pg/mL) (Figure 4A). The levels of proinflammatory cytokines such as IL-1**β**, IL-6, and TNF-α known to synergize with RANK-L, were assessed [15]. The expression level of IL-1β and IL-6 remained below the threshold of detection in MF supernatants even after HIV-1 infection, and the level of TNF-α mRNA was not modified upon infection (Appendix A). Second, we investigated whether HIV-infected MF could, by a bystander effect, promote OC differentiation and bone degradation. To this end, OC precursors were differentiated in the presence of CmHIV or CmNI, and the fusion index and bone degradation were measured. We checked that treatment of OC precursors with CmHIV did not induce cell death compared to CmNI by measuring cell density (Appendix A) and analyzing morphological features of nuclei (area, circularity and solidity). As infection of OC precursors promotes their differentiation [25], OC precursors were pre-treated, before addition of conditioned media, with Maraviroc to prevent HIV-1 entry. The fusion of OC precursors and bone degradation were not significantly increased by incubation with CmHIV compared to CmNI (Figure 4B). Finally, we tested whether CmHIV could impact the migration of OC precursors. The recruitment of OC precursors from blood to bones requires proteases in vivo [44], and we showed previously that defects in the 3-dimensional (3D) protease-dependent mesenchymal migration of these cells in vitro correlates with lower recruitment of OC to bones in vivo [25,45]. In Matrigel, human myeloid precursors use the mesenchymal migration [25,45,46,47]. We found that, using CmHIV as chemoattractant in the lower chamber, the percentage of cells infiltrating Matrigel was enhanced (Figure 4C). This is not due to a potential infection of the cells inside the matrix by viral particles contained in CmHIV as addition of viral stock in the lower chamber has no effect on OC precursor migration. As shown in Figure 4D, we also found an increase of the percentage of migrating OC precursors using as chemoattractant a range of recombinant RANK-L concentrations close to the ones measured in CmHIV. As a proof of concept that the migration enhanced by CmHIV was RANK-L-dependent, we pre-treated MF supernatants with 100ng/mL of OPG, an antagonistic endogenous receptor of RANK-L. While OPG treatment has no or little effect on CmNI-exposed cells, it was sufficient to almost abolish the CmHIV-induced increase in OC precursor migration (Figure 4C). 

In conclusion, CmHIV is not sufficient to promote differentiation of OC precursors in vitro, but it promotes their 3D mesenchymal migration, in particular through RANK-L secretion, suggesting that HIV-1-infected MF could exert a bystander effect to favor active recruitment of OC precursors to bones.

## 3. Discussion

The mechanisms underlying bone defects and exacerbation of osteolysis resulting from HIV-1 infection need to be further investigated. It is established that different MF subsets are present in bones. They are essential for the maintenance of the bone architecture notably during tissue regeneration and inflammatory responses [32,34,35,48,49]. Here, we studied whether MF could participate into osteolysis associated with HIV infection. HIV-1-infected human MF acquired some OC features but failed to resorb bone. Importantly, we report that infection of MF enhanced their RANK-L secretion that promotes OC precursor recruitment. Thus, in addition to the multiple factors including antiretroviral treatment [2] that contribute to bone loss in HIV-1-infected patients, we described here a novel mechanism by which the virus favors bone resorption. 

Upon HIV-1 infection with different MF-tropic viral strains, MF acquired OC characteristics, such as multinucleation and ability to organize podosomes into circular structures, when seeded on glass [39] or bone (this study). The effect on fusion was also observed using transmitted/founder viral strains, which are single variants responsible for the viral transmission in patients. In addition, an increase in RhoE and β3 integrin levels, two proteins essential for OC podosome patterning into a functional sealing zone [14,18], was detected in infected MF. When plated on bone matrices, infected MF organized their actin cytoskeleton into circular and vinculin-positive superstructures reminiscent of sealing zone. These superstructures were not present in non-infected MF (Appendix A). The F-actin structures formed in HIV-1-infected MF promoted strong adhesion to the substrates but produced only superficial and limited abrasion of bone substrates. These sealing zone-like structures are smaller in size, resembling the podosome rosettes induced by activated Src kinases in fibroblasts [50,51,52]. Thus, in HIV-1-infected MF, the induction of a circular podosome organization associated with an increase in RhoE and β3 integrin levels are not sufficient for generating functional sealing zone. It is likely that they are unable to acidify mineral, as we have previously shown that HIV-1 infection increases the capacity of MF to degrade collagen matrices in 2D and in 3D [30]. In agreement with the inability of these structures to degrade bone, we did not notice any increase in the mRNA level of NFATc-1, the OC-master transcription factor [53,54] and several genes under its control such as TRAP and Ctsk. Mature dendritic cells are able, in some specific conditions, to trans-differentiate into bone resorbing multinucleated OC [36,37,38]. This is not the case for fully differentiated MF in the context of HIV-1 infection. However, knowing the plasticity of hematopoietic progenitors, it would be interesting to consider this possibility at earlier stages of MF differentiation. 

As the major stimulus for osteoclastogenesis is RANK-L [15], we then determined whether HIV-1-infected MF could be an active source of soluble RANK-L and thus potentially exert a bystander effect on recruitment and/or fusion of OC precursors. In CmHIV (supernatant of infected MF), we detected a three-fold increase of RANK-L concentration compared to CmNI (supernatant of non-infected MF). This concentration is equivalent to the one produced by activated T cells, which are known as efficient RANK-L producers [55]. Although MF were not described to be a classical source of RANK-L, in situ hybridization have previously reported RANK-L expression in MF in inflammatory contexts such as periodontal diseases [56,57]. Increase in RANK-L secretion by HIV-1-infected MF could result from either an increased production or a cleavage of the membrane-associated forms [58]. Thus, we propose that, in addition to osteoblasts, osteocytes, T and B lymphocytes, infected MF can be a novel source of RANK-L. Other studies have already proposed that HIV-1 alters the secretion of osteoclastogenic regulatory factors. In HIV-transgenic rats and in sera from HIV-1 infected individuals, a marked increase in RANK-L production together with a reduction in OPG production by lymphocytes has been reported [22]. In both models, the increase of RANK-L/OPG ratio was correlated with a marked reduction in bone mineral density [6,21,22]. In contrast to RANK-L, the levels of the inflammatory cytokines (IL-1β, IL-6, and TNF-α described as cooperative factors of RANK-L-dependent osteoclastogenesis) remained undetectable in infected MF. 

Although CmHIV notably contained RANK-L, we showed in vitro that incubation of OC precursors with CmHIV could not efficiently promote their multinucleation. However, it stimulated their migration in 3D Matrigel matrices. It is likely that the average concentration of RANK-L secreted in the CmHIV (300 pg/mL) was under the threshold for triggering multinucleation of OC precursors in vitro. Actually, it is much lower than the one used to differentiate monocytes in OC in vitro (about 2 logs above). However, it is reasonable to propose that the increase in RANK-L secretion by infected MF could contribute to enhanced osteolysis in vivo by favoring the recruitment of OC precursors to bones. Actually, we showed that the mesenchymal 3D migration of OC precursors is increased using, as chemo-attractant, recombinant RANK-L at concentrations equivalent to the ones of CmHIV (range 100–1000 pg/mL). In CmHIV, RANK-L mainly contributes to enhanced migration of OC precursors, as CmHIV-induced increase of OC precursor migration was counteracted by OPG. However, it was not excluded that other cytokines could participate in this induction [59]. We have reported that a correlation exists between enhanced mesenchymal migration of OC precursors and the number of OC in bones [45]. Therefore, we expect that the enhancement of OC precursor migration by the microenvironment generated by HIV-1-infected MF could favor OC recruitment to bones, and thus participate in bone disorders observed in infected patients. In accordance with this proposal, HIV-1-transgenic rats have a reduced bone mass as a consequence of an increased number of OC [22]. Furthermore, transgenic mice expressing the viral protein Nef, a key factor in the bone resorption activity of HIV-1-infected OC, exhibited a reduced bone density and a marked increase in OC compared with controls [25]. In conclusion, in addition to the mechanisms described previously by us and others [6,21,25], this study identifies a novel mechanism (i.e., RANK-L secretion by HIV-1-infected MF) which likely participates to the high osteolysis associated with HIV-1 infection. 

## 4. Material and Methods

### 4.1. Preparation of Monocyte-Derived Macrophages and Monocyte-Derived Osteoclasts

Buffy coats were provided by Etablissement Français du Sang, Toulouse, France, under contract 21/PLER/TOU/IPBS01/2013-0042. According to articles L1243-4 and R1243-61 of the French Public Health Code, the contract was approved by the French Ministry of Science and Technology (agreement number AC 2009-921, Jan 2017). Written informed consent was obtained from all participants. Monocytes from healthy subjects were isolated from buffy coats, seeded on glass coverslips (Marienfeld, Lauda-Königshofen, Germany) in 6-well plates (1.5 × 10^6^ cells/well) or 24-well plates (5 × 10^5^ cells/well) or bone slices (ImmunoDiagnostic Systems, Boldon, UK) in 96-well plates (5 × 10^4^ cells/well) and differentiated into macrophages and osteoclasts (OC) as previously described [25]. Briefly, to obtain MF, purified CD14^+^ monocytes were differentiated for 7 days in RPMI-1640 medium (GIBCO) supplemented with 10% fetal bovine serum (FBS, Sigma-Aldrich, St. Louis, MO, USA), L-glutamin (10 mM, Gibco, Grand Island, NY, USA), penicillin-streptomycin (1%, Gibco), and human recombinant M-CSF (20 ng/mL, PeproTech, Rocky Hill, NJ, USA). For OC differentiation, monocytes were cultured in complete medium supplemented with M-CSF (50 ng/mL) and RANK-L (30 ng/mL, Miltenyi Biotech, Bergisch Gladbach, Germany). The medium was replaced every 3 days with medium containing M-CSF (25 ng/mL) and RANK-L (100 ng/mL). MF and OC from the same donor were used. For migration experiments, OC precursors at day 3 of differentiation were used as in [25]. For differentiation of OC precursors by conditioned medium, monocytes were pre-exposed to sub-concentration of RANK-L (10 ng/mL) for three days, then were cultured in the presence of CmNI (conditioned medium of uninfected control MF) or CmHIV (conditioned medium of HIV-1-infected MF) supplemented with FBS (10%) during additional nine days. To prevent infection of precursors, Maraviroc (5 µM, Sigma-Aldrich, Saint-Louis, MO, USA) was added to the culture medium 30 min before adding conditioned medium.

### 4.2. Viruses, HIV-1 Infection, and Preparation of Conditioned Media

Proviral infectious clones of the MF-tropic HIV-1 (ADA and NLAD8) were kindly provided by Serge Benichou (Institut Cochin, Paris, France) while Transmitted /Founder strains (SUMA, cat# 11748 and THRO, cat# 11745) were obtained through the NIH AIDS Reagent Program, Division of AIDS, NIAID, NIH from Dr. John Kappes and Dr. Christina Ochsenbauer. Virions were produced by transient transfection of 293T cells with proviral plasmids, as previously described [30]. HIV-1 p24 antigen concentration of viral stocks was assessed by a home-made ELISA (see below). HIV-1 infectious units were quantified, as reported [60] using TZM-bl cells (Cat#8129, NIH AIDS Reagent Program, Division of AIDS, NIAID, NIH from Dr. John C. Kappes, and Dr. Xiaoyun Wu). MF and OC were infected at MOI 0.5 (corresponding to 0.7 ng of p24 for 2 × 10^6^ cells). HIV-1 infection and replication were assessed 10 days post-infection by measuring the expression of Gag gene by RT-qPCR, the infection index calculated after Gag immunostaining and the p24 level in cell supernatants by ELISA.

For preparation of conditioned media (CmHIV), MF were infected with HIV-1 (NLAD8 strain) at a MOI of 0.5. The conditioned control medium (CmNI) was obtained from non-infected MF from the same donor. After washing at day 1 post infection, culture media were collected at least at day 10 post infection, centrifuged to eliminate cell debris, and aliquots were stored at −80 °C.

### 4.3. RNA Extraction and qRT-PCR

Total RNA was extracted using ready-to-use TRIzol Reagent (Ambion, Life Technologies, Austin, TX, USA) and purified with RNeasy Mini kit (Qiagen, Germantown, MD, USA) following manufacturer’s instructions. Complementary DNA was reverse transcribed from 1 μg total RNA with Moloney murine leukemia virus reverse transcriptase (Sigma-Aldrich) using dNTP (Promega, Madison, WI, USA) and random hexamer oligonucleotides (ThermoFisher, Waltham, MA, USA) for priming. qPCR was performed using SYBR green Supermix (OZYME, Saint-Cyr-l’École, France) in an ABI7500 Prism SDS real-time PCR detection system (Applied Biosystems, Foster City, CA, USA). The mRNA content was normalized to β-actin mRNA and quantified using the 2^−∆∆*C*t^ method. Primers used for cDNA amplification were purchased from Sigma-Aldrich and are listed in Appendix A.

### 4.4. ELISA

Cytokine quantification was performed in cell supernatants by sandwich ELISA using RANK-L kits (ELISAGenie, London, UK), IL-6 kits (BD Biosciences Franklin Lakes, NJ, USA), or IL-1β kits (Invitrogen, Waltham, MA, USA) according to manufacturer’s instructions. Antibodies used are described in Appendix A. HIV-1 p24 concentration of viral stocks and p24 released by infected MF was measured by a previously described home-made sandwich ELISA [60] with NIH reagents, NIH AIDS Reagent Program, Division of AIDS, NIAID.

### 4.5. Immunofluorescence Microscopy

Immunofluorescence (IF) experiments were performed as described (6, 16). Briefly, cells were fixed with PFA 3.7% (Sigma-Aldrich), sucrose 30 mM in PBS (Gibco), permeabilized with Triton X-100 0.3% (Sigma-Aldrich) for 10 min, and saturated with PBS BSA 1% (Euromedex, Souffelweyersheim, France ) for 30 min. Cells were incubated with primary antibodies diluted in PBS BSA 1% for 1 h, washed and then incubated with corresponding secondary antibodies (LifeTechnologies, Carlsbad, CA, USA), AlexaFluor488- or TexasRed-labeled phalloidin (Invitrogen) and DAPI (Sigma-Aldrich) in PBS BSA 1% for 30 min. Coverslips were mounted on a glass slide using fluorescence mounting medium (Dako, Santa Clara, CA, USA). Slides were visualized with a Leica DM-RB fluorescence microscope (Leica Microsystems, Wetzlar, Germany) or on a FV1000 confocal microscope (Olympus, Tokyo, Japan). Images were processed with ImageJ and Adobe Photoshop softwares. Primary antibodies used are described in Appendix A. The HIV infection index (total number of nuclei in HIV-stained cells divided by total number of nuclei × 100) and fusion index (total number of nuclei in multinucleated cells divided by total number of nuclei × 100) were quantified, as in [39]. The number of cells with podosomes, podosome ring structures, and sealing zones were quantified after phalloidin and vinculin staining. For conditioned media treatment of OC precursors, cell viability was assessed by measuring cell density (number of nuclei per 1000 µm²), and morphological features of nuclei such as mean area, circularity (4πarea/perimeter), and solidity (area/convex area). At least 200 nuclei were analyzed per image.

### 4.6. Scanning Electron Microscopy 

Scanning electron microscopy observations were performed as previously described [25]. Briefly, following complete cell removal by immersion in water and scraping, bone slices were dehydrated in a series of increasing ethanol. Critical point was dried using carbon dioxide in a Leica EMCPD300. After coating with gold, bone slices were examined with a FEI Quanta FEG250 scanning electron microscope.

### 4.7. Immunoblot Analyses

Total protein lysates were extracted as previously described [61]. Total proteins were separated through SDS-polyacrylamide gel electrophoresis, transferred, and immunoblotted overnight at 4 °C with indicated primary antibodies. Antibodies used are described in Appendix A. Proteins were visualized with Amersham ECL Prime Western Blotting Detection Reagent (GE Healthcare, Chicago, IL, USA). Chemiluminescence was detected with ChemiDoc Touch Imaging System (Bio-Rad Laboratories, Hercules, CA, USA). Quantification of immunoblot intensity was performed using Image Lab (BioRad. Quantifications were normalized to tubulin.

### 4.8. 3D Migration Assays

3D migration assays of OC precursors in Matrigel (10–12 mg/mL, BD Biosciences) were performed as described [25]. Recombinant RANK-L was used as chemoattractant in the lower chamber at the indicated concentrations. Supernatant of infected (CmHIV) or uninfected (CmNI) macrophage, incubated or not with recombinant OPG (100 ng/mL, Miltenyi Biotech), was supplemented with 20% FBS and added in the lower chamber for 3D migration assay. OC precursors were detached and resuspended in RPMI supplemented with 0.5% FBS. Total of 5 × 10^4^ cells were added at the top of each well and let to migrate for 48 h or 72 h. Cells were then fixed in 3.7% PFA for 1 h and pictures were taken automatically with a 10× objective at constant intervals (z = 30 µm) using the motorized stage of an inverted microscope (Leica DMIRB, Leica Microsystems). Cells were counted using ImageJ software.

### 4.9. Bone Resorption Assays 

To assess bone resorption activity, monocytes were seeded on bovine cortical bone slices (ImmunoDiagnostic Systems) and differentiated into MF or OC. Following complete cell removal by several washes with water, bone slices were stained with toluidin blue (Sigma-Aldrich) to detect resorption pits under a light microscope (Leica DMIRB, Leica Microsystems). Surface of bone degradation areas were quantified manually with ImageJ software. Cross-linked C-telopeptide collagen I (CTX) concentrations were measured using betaCrosslaps assay (ImmunoDiagnostic System) in the culture medium of OC grown on bone slices [25]. 

### 4.10. Statistical Analysis

The exact values of n (donors) can be found in the figure legends. All statistical analyses were performed using GraphPad Prism 6.0 (GraphPad Software Inc., San Diego, CA, USA). The statistical tests were chosen according to the following. Two-tailed paired or unpaired t-test was applied on data sets with a normal distribution (determined using Kolmogorov-Smirnov test), whereas two-tailed Mann-Whitney (unpaired test) or Wilcoxon matched-paired signed rank tests were used otherwise. *p* < 0.05 was considered as the level of statistical significance (* *p* ≤ 0.05; ** *p* ≤ 0.01; *** *p* ≤ 0.001; **** *p* ≤ 0.0001).

## 5. Conclusions

In bone, both MF derived from infiltrating monocytes and resident MF participate to tissue homeostasis [49]. It is conceivable that, like OC and OC precursors [25], MF or their precursors could be infected in the bone environment and, by secreting RANK-L, contribute to a local supportive environment for osteoclastogenesis and, possibly, to an increase of circulating RANK-L. We propose that the virus disrupts the RANK-L/OPG equilibrium by stimulating the production of RANK-L by T and B lymphocytes [6,21,22] and also by MF (this study). To conclude, using a RANK-L-dependent bystander mechanism, HIV-1-infected MF participate in the recruitment and differentiation of OC precursors that participate in bone disorders encountered in HIV-1^+^ patients. 

## Figures and Tables

**Figure 1 ijms-21-03154-f001:**
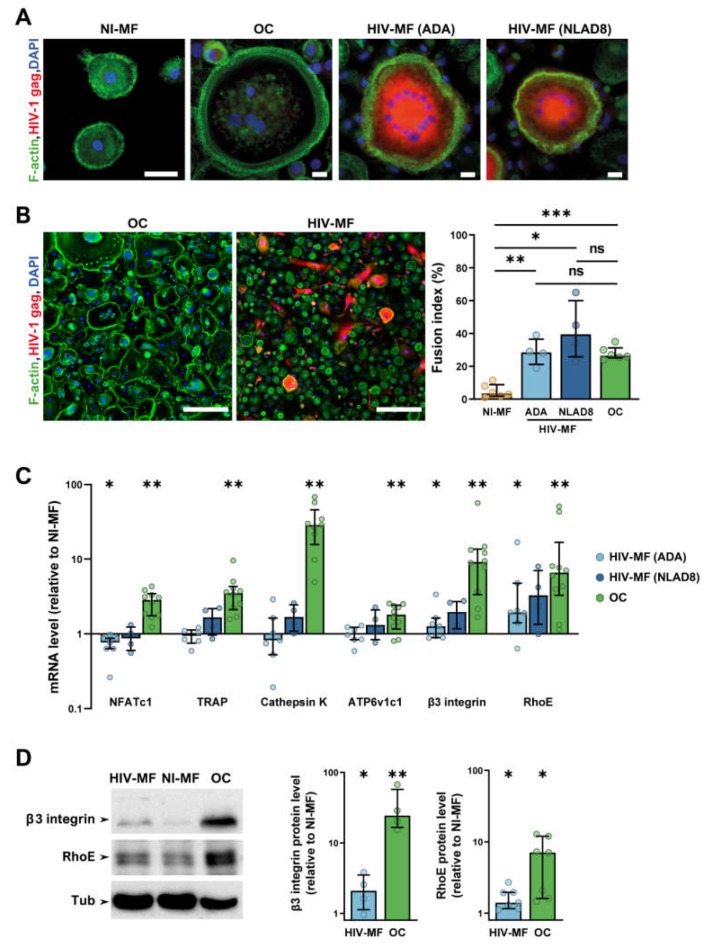
HIV-1 infection induces macrophage (MF) fusion and the expression of some osteoclast (OC) markers. MF were infected or not with HIV-1 (indicated strains), and compared to autologous OC for cell fusion (**A**,**B**) and expression of different OC markers (**C**,**D**). (**A**) Representative immunofluorescence (IF) images of uninfected MF (NI-MF), MF infected with HIV-1 (HIV-MF, ADA, or NLAD8 strain), and OC after staining for HIV-1 gag (red), F-actin (green), and nuclei (DAPI, blue). Scale bar, 10 µm. (**B**) Left panels: Representative large IF images of HIV-MF (ADA strain) and OC after staining for F-actin (green), HIV-1 gag (red), and nuclei (DAPI, blue). Scale bar, 200 µm. Right panel: Quantification of the fusion index evaluated by IF, corresponding to the percentage of nuclei within multinucleated cells. Histograms represent median and error bars are interquartile range, *n* = 4 to 6 donors, 300 cells analyzed per donor and per condition. (**C**) mRNA expression of genes overexpressed in OC measured by RT-PCR using the ΔΔC_T_ method in HIV-MF (ADA or NLAD8 strain) and in OC. Actin mRNA level was used as control. Values are normalized to mRNA level in autologous uninfected MF. Histograms represent median and error bars are interquartile range, *n* = 4 to 10 donors. (**D**) NI-MF, HIV-MF (NLAD8 strain), and OC lysates were subjected to Western blot using antibodies against β3-integrin, RhoE, and α-Tubulin as loading control. A representative blot (left panel) and quantification of the protein level ratio over autologous NI-MF (right panel) are shown. Histograms represent median and error bars are interquartile range, *n* = 4 to 7 donors. * *p* ≤ 0.05; ** *p* ≤ 0.01; *** *p* ≤ 0.001, ns: not significantly different.

**Figure 2 ijms-21-03154-f002:**
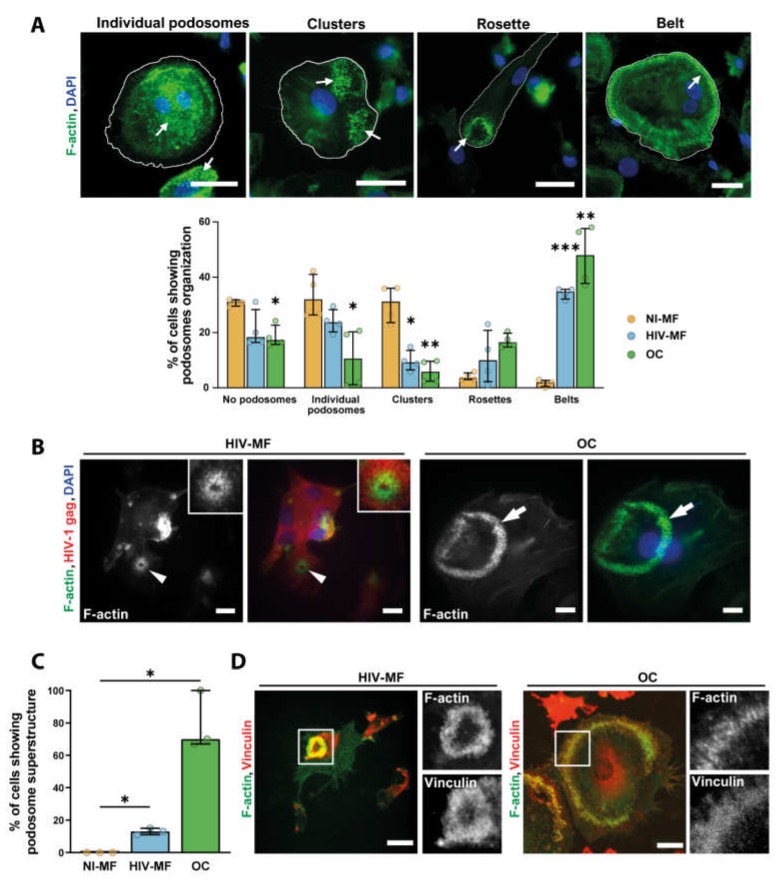
HIV-1 infection of MF promotes podosome organization into super-structures. (**A**) Top: Representative IF images of different podosome organizations in MF or OC seeded on glass coverslips, after staining for F-actin (green) and nuclei (DAPI, blue). White arrows show podosome structures. Scale bar, 20 µm. Bottom: Quantification of the different podosome organizations in non-infected MF (NI-MF), HIV-1-infected MF (HIV-MF, ADA strain), and OC represented in percentage of total cells. Histograms represent median and error bars are in interquartile range, *n* = 4 donors, 300 cells analyzed per donor. (**B**) Monocytes were seeded on bone slices and differentiated into MF or OC. Then, at day 7, MF were infected with HIV-1 (NLAD8 strain) and all cells were fixed at day 14. Left: representative IF images of podosome super structures in HIV-MF (arrowheads) and OC (arrows) stained for HIV-1 gag (red), F-actin (green), and nuclei (DAPI, blue). Scale bar, 5 µm. Inserts show two-fold magnification of the podosome structure in HIV-MF. (**C**) Quantification of the percentage of MGC showing podosome organization on bones, evaluated by IF. Histograms represent median and error bars are in interquartile range, *n* = 3 donors, 100 cells analyzed per condition. (**D**) Representative image of podosome super-structures formed by HIV-MF (NLAD8 strain, left) or OC (right) seeded on bone slices, after staining for F-actin (green) and vinculin (red). Scale bar, 10 µm. Inserts show two-fold magnification of white square. * *p* ≤ 0.05; ** *p* ≤ 0.01; *** *p* ≤ 0.001.

**Figure 3 ijms-21-03154-f003:**
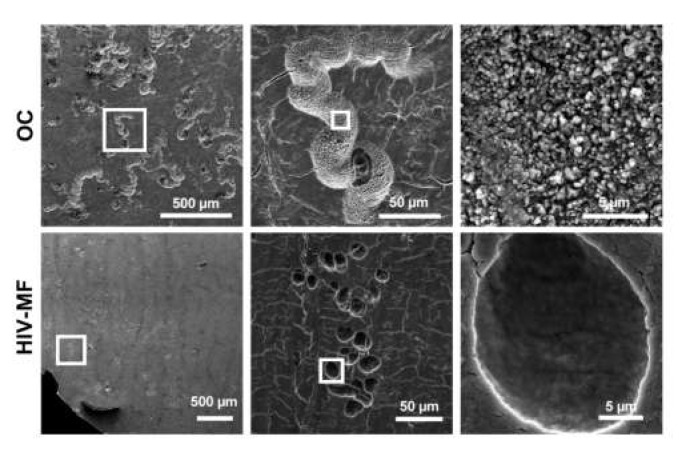
Superficial bone resorption by HIV-1-infected MF compared to OC. Monocytes were seeded on bone slices and differentiated into MF or OC. At day 7, MF were infected with HIV-1. At day 14, cells were removed and bone slices were stained with toluidine blue. Representative scanning electron microscopy images showing bone resorption pits formed by OC or infected MF (HIV-MF, NLAD8 strain). Scale bars indicated.

**Figure 4 ijms-21-03154-f004:**
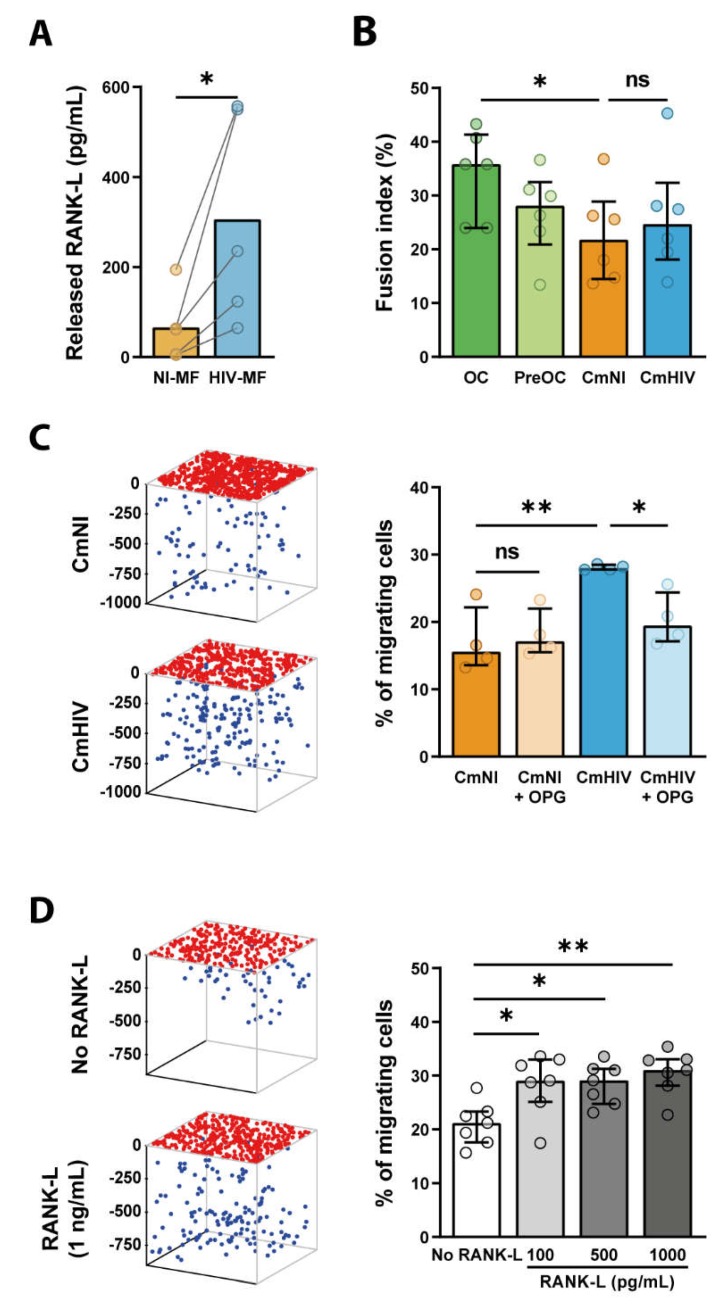
RANK-L is secreted by HIV-1-infected MF and promotes 3D migration of OC precursors. (**A**) MF were infected with HIV-1 (HIV-MF, NLAD8 strain) for 7 days, and the level of released RANK-L was measured by ELISA in the supernatant and compared to uninfected MF (NI-MF) from the same donor. *n* = 5. (**B**) Monocytes were differentiated for 3 days in presence of sub-optimal concentrations of RANK-L and then exposed to the supernatant of infected (CmHIV) or uninfected (CmNI) MF for additional 10 days. Cells were then fixed and the fusion index was quantified by IF. Bars represent median, *n* = 6 donors. (**C**, **D**) CmHIV or CmNI pre-treated or not with OPG (100ng/mL) (**C**), or different concentrations of recombinant RANK-L (**D**) was added at the bottom of 3D Matrigel matrices. OC precursors were then seeded at the top of the matrices and allowed to migrate. Left: 3D representation of the positions of OC precursors that have migrated (blue dots) or not (red dots) in a representative migration assay. Ticks represent depth in microns. Right: quantification of the percentage of cells inside the matrix. (**C**) *n* = 4 donors; (**D**) *n* = 7 donors, performed in triplicates. Histograms represent median and error bars are interquartile range. * *p* ≤ 0.05; ** *p* ≤ 0.001, ns: not significantly different.

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
