# Peer review of "HIV-1-Infected Human Macrophages, by Secreting RANK-L, Contribute to Enhanced Osteoclast Recruitment"

_ijms, 2020, doi:10.3390/ijms21093154_

Round 1

Reviewer 1 Report

The authors investigated the role of HIV-infected macrophages in the osteoclast regulation. They showed that infected macrophages are not able to differentiate into osteoclasts but could have a role in the regulation of osteoclast precursors recruitment.

The paper should be improved.

Lines 47-48: Remove “that fuse with embryonic resident long-lived OC”

Line 58: TRAP is not a resorption-related enzyme

Lines 107-110: The insertion is not exhaustive; They should measure and show the number of cells with X nuclei. For example, they should count the cells with 3-5 nuclei, cells with 5-10 nuclei and more than 10 nuclei.

Figure 2A-C: Insert statistical analysis

Figure 3A-B-C: Remove from the main text and insert in the supplementary figures

Line 224: Show the data regarding cell death

All figures must have the same style

Author Response

Lines 47-48: Remove “that fuse with embryonic resident long-lived OC”

Line 58: TRAP is not a resorption-related enzyme

We have modified these sentences, see lines 46-47 and 56-58.

Lines 107-110: The insertion is not exhaustive; They should measure and show the number of cells with X nuclei. For example, they should count the cells with 3-5 nuclei, cells with 5-10 nuclei and more than 10 nuclei.

As requested, we have included the results on the number of nuclei per cell for HIV-MGC and OC in the result section, lines 107-108.

Figure 2A-C: Insert statistical analysis

We thank the reviewer for noticing these omissions, which have been corrected in the revised Figure 2A and 2C.

Figure 3A-B-C: Remove from the main text and insert in the supplementary figures

It has been done. We created a new Supplemental Figure 3A-C, and previous Supplemental Figure 3 is now Supplemental Figure 4A-B.

Line 224: Show the data regarding cell death

As requested by the reviewer, the data regarding cell death have been added as Supplemental Figure 4C. Cell viability has been measured by quantification of the number of nuclei per 1000 µm2 (cell density) which is similar between CmHIV-treated cells and CmNI-treated cells (See new Supplemental Figure 4C and methods lines 361-362).

All figures must have the same style

All the Figures and Supplemental Figures have now the same style in term of histogram presentation, please note in particular the changes made in Figure 2A.

Reviewer 2 Report

The authors have addressed all my comments and questions about the manuscript satisfactorily.

Author Response

N/A

Round 2

Reviewer 1 Report

The authors did not completely reply to my previous review.

They can not measure the effects on cell death measuring the number of nuclei per surface. They should use Annexin V/ PI protocol or other apoptosis detection kits.

Round 3

Reviewer 1 Report

The authors added further analysis regarding cell viability

This manuscript is a resubmission of an earlier submission. The following is a list of the peer review reports and author responses from that submission.

Round 1

Reviewer 1 Report

The authors described how HIV-1-infected macrophages play a role in bone loss observed in patients. The paper could be interesting. However, there are some issues that should be clarified:

Line 43: embryonic osteoclasts? Line 101: "fusion index of HIV-1-infected MF was similar to the one of OC (monocytes differentiated with M-102 CSF and RANKL from the same donor) (Fig. 1A-B)". DAPI staining they showed in Figure 1 is not representative of the fusion index since few nuclei are observed in Osteoclasts. In the text, they mentioned that they found an increase of ITGB3 e Rhoe expression in ADA and NLDA8 infected cells. In the Figure 1C, they showed a statistically significant result only for ADA infected cells. Please correct the sentence. In Figure 1D, please insert the asterisks Regarding the analysis on the conditioned medium, they should measure osteoprotegerin. This could be important to understand the free available RANKL. To demonstrate that RANKL is the main player of OC precursor migration, they should counteract RANKL action by incubation with neutralizing monoclonal antibody and analyze the migration. Editing process is required (for example Line 50: Correct “IL-17 favor tosteoclastogenesis”, etc)

Reviewer 2 Report

Mascarau et al have examined that HIV-1-infected human macrophages, by secreting RANKL, contribute to enhanced osteoclastogenesis. Overall this is a well presented paper. The methods are clearly set out the results are clear and concise. However, there are some parts of the manuscript that need some clarification.

In the introduction the authors examine the role of osteoclasts in bone loss in HIV patients. Could they also add other extenuating factors such as the effects of HIV on osteoblasts and osteocytes, and mesenchymal stem cells. Does a lack of mechanical loading of bone due to HIV patient’s loss of muscle tone contribute to the loss of bone in these HIV patients irrespective of osteoclast activity?

Although the authors have shown a 3D migratory enhancement of OC precursors using the Matrigel assay they have not shown evidence of osteoclast activity via RANKL secretion so is the title correct that secreted RANKL has enhanced osteoclastogenesis?

In figure 4 there seems to be some variation in the donors used n=5 to 8 donors, n=6-10 donors  and n= 3 to 5 donors… Why was this the case and are the results comparable between the varying assays in Figure 4? Also were the donors from a specific sex. Is there any data to compare the variation of using same sex samples?

In the migration studies induced by RANKL by the HIV infected CmHIV did the authors add exogenous osteoprotegerin to inhibit the potential RANKL affects to confirm that RANKL is causing this effect?

In figure 3B there is quite a marked variation in the degradation index of OC. Could the authors comment as to why there is such a large variation.

What were the cell seeding densities onto the bovine cortical bone slice resorption pit assays?

In the migration assay what were the levels of cell viability of the migrated cells and the cells on top of the Matrigel over the period of culture. In addition are the macrophage cells 100% viable after HIV infection? Does the viability start to reduce with increasing culture times?  

Could the authors comment in the discussion as to the effect of combinational retroviral therapy and its enhanced affect of bone loss (Moran et al 2016) and how this would fit into their hypothesis/results of this manuscript. Also, a recent publication by Raynaud-Messina et al, 2018 reported that osteoclasts are host target cells for HIV-1 that become more osteolytic as a consequence of larger and more degradative Sealing Zones and that the viral protein Nef is a key regulator to this mechanism. Does this correlate with your findings as the infected MF showed smaller sealing-zone structures and limited bone resorption?

Minor

Line 42  “While osteoblasts develop from cells of mesenchymal origin,”  “…of a mesenchymal origin”

Line 50 spelling error “tosteoclastogenesis”

Line 177 – should it be “3 donors out of 5”

Line 271 Remove reference information  “#2706;Wakkach, 2008 #2931}”

Mascarau et al have examined that HIV-1-infected human macrophages, by secreting RANKL, contribute to enhanced osteoclastogenesis. Overall this is a well presented paper. The methods are clearly set out the results are clear and concise. However, there are some parts of the manuscript that need some clarification.

In the introduction the authors examine the role of osteoclasts in bone loss in HIV patients. Could they also add other extenuating factors such as the effects of HIV on osteoblasts and osteocytes, and mesenchymal stem cells. Does a lack of mechanical loading of bone due to HIV patient’s loss of muscle tone contribute to the loss of bone in these HIV patients irrespective of osteoclast activity?

Although the authors have shown a 3D migratory enhancement of OC precursors using the Matrigel assay they have not shown evidence of osteoclast activity via RANKL secretion so is the title correct that secreted RANKL has enhanced osteoclastogenesis?

In figure 4 there seems to be some variation in the donors used n=5 to 8 donors, n=6-10 donors  and n= 3 to 5 donors… Why was this the case and are the results comparable between the varying assays in Figure 4? Also were the donors from a specific sex. Is there any data to compare the variation of using same sex samples?

In the migration studies induced by RANKL by the HIV infected CmHIV did the authors add exogenous osteoprotegerin to inhibit the potential RANKL affects to confirm that RANKL is causing this effect?

In figure 3B there is quite a marked variation in the degradation index of OC. Could the authors comment as to why there is such a large variation.

What were the cell seeding densities onto the bovine cortical bone slice resorption pit assays?

In the migration assay what were the levels of cell viability of the migrated cells and the cells on top of the Matrigel over the period of culture. In addition are the macrophage cells 100% viable after HIV infection? Does the viability start to reduce with increasing culture times?  

Could the authors comment in the discussion as to the effect of combinational retroviral therapy and its enhanced affect of bone loss (Moran et al 2016) and how this would fit into their hypothesis/results of this manuscript. Also, a recent publication by Raynaud-Messina et al, 2018 reported that osteoclasts are host target cells for HIV-1 that become more osteolytic as a consequence of larger and more degradative Sealing Zones and that the viral protein Nef is a key regulator to this mechanism. Does this correlate with your findings as the infected MF showed smaller sealing-zone structures and limited bone resorption?

Minor

Line 42  “While osteoblasts develop from cells of mesenchymal origin,”  “…of a mesenchymal origin”

Line 50 spelling error “tosteoclastogenesis”

Line 177 – should it be “3 donors out of 5”

Line 271 Remove reference information  “#2706;Wakkach, 2008 #2931}”